# Perspectives of VA healthcare from rural women veterans not enrolled in or using VA healthcare

Carly M. Rohs[1,2¤]*, Karen R. Albright[1,3], Lindsey L. Monteith[2,4], Amber D. Lane[1], Kelty B. Fehling[1]

1 Seattle-Denver Center for Innovation (COIN), Rocky Mountain Regional VA Medical Center, Aurora, Colorado, United States of America, 2 VA Rocky Mountain Mental Illness, Research, Education and Clinical Center (MIRECC) for Suicide Prevention, Aurora, Colorado, United States of America, 3 Division of General Internal Medicine, Department of Medicine, University of Colorado Anschutz Medical Campus, Aurora, Colorado, United States of America, 4 Department of Physical Medicine and Rehabilitation, University of Colorado Anschutz Medical Campus, Aurora, Colorado, United States of America

¤ Current address: Rocky Mountain Mental Illness Research, Education and Clinical Center for Suicide Prevention, Rocky Mountain Region VA Medical Center, Aurora, CO, United States of America
* carly.rohs@va.gov

**Data Availability Statement:** Institutional restrictions prohibit us from sharing this data publicly. Specifically, we do not have approval by

## Abstract

### Purpose

Women Veterans have unique healthcare needs and often experience comorbid health conditions. Despite this, many women Veterans are not enrolled in the Veterans Health Administration (VHA) and do not use VHA services. Underutilization of VHA services may be particularly prevalent among rural women Veterans, who may experience unique barriers to using VHA care. Nonetheless, knowledge of rural women Veterans and their experiences remains limited. We sought to understand rural women Veterans' perceptions and needs related to VHA healthcare, including barriers to enrolling in and using VHA services, and perspectives on how to communicate with rural women Veterans about VHA services.

### Methods

Rural women Veterans were recruited through community engagement with established partners and a mass mailing to rural women Veterans not enrolled in or using VHA healthcare. Ten virtual focus groups were conducted with a total of twenty-nine rural women Veterans (27 not enrolled in VHA care and 2 who had not used VHA care in the past 5 years) in 2021. A thematic inductive analytic approach was used to analyze focus group transcripts.

### Findings

Primary themes regarding rural women Veterans' perceptions of barriers to enrollment and use of VHA healthcare included: (1) poor communication about eligibility and the process of enrollment; (2) belief that VHA does not offer sufficient women's healthcare services; and (3) inconvenience of accessing VHA facilities.

our regulatory authority (the VA Rocky Mountain Regional VA Research and Development Committee) to share de-identified data publicly for this study. Rather, de-identified data can be accessed with a Data Use Agreement and verification of IRB approval from the requestor. We have provided contact information for our local regulatory authority (the VA Rocky Mountain Regional VA Research & Development Committee) to review any such requests. Contact name: Brandi Lippman Phone Number: 720-857-5106 E-mail: vhaechresearchadmin@va.gov.

**Funding:** KF received funding from Veteran Health Administration's Office of Women's Health. This was funding received for an evaluation project, where there was no award number. The funder did not play a role in the study design, data collection and analysis or preparation of the manuscript. They did contribute to the agreement to publish findings. https://www.womenshealth.va.gov/.

**Competing interests:** The authors have declared that no competing interests exist.

## Conclusion

Although VHA has substantially expanded healthcare services for women Veterans, awareness of such services and the nuances of eligibility and enrollment remains an impediment to enrolling in and using VHA healthcare among rural women Veterans. Recommended strategies include targeted communication with rural women Veterans not enrolled in VHA care to increase their awareness of the enrollment process, eligibility, and expansion of women's healthcare services. Creative strategies to address access and transportation barriers in rural locations are also needed.

## Introduction

During their military service, many women experience stressful and traumatic experiences, such as military sexual trauma, combat, separation from family and friends, and gender bias [1–3]. These experiences are associated with elevated risk for health concerns, such as chronic pain, depression, substance use disorders, and suicidal ideation [4–8]. Further, studies have noted high rates of comorbid health conditions among women Veterans [9]. As such, it is essential that women Veterans have access to the comprehensive healthcare services provided by the Veterans Health Administration (VHA), which has made concerted efforts to address women Veterans' specific healthcare needs through women's health clinics and Patient Aligned Care Teams that work closely with Women Veterans Program Managers (WVPM) in the provision of reproductive healthcare and maternity care coordination [10–12]. The VHA has committed to having a WVPM located in every regional office, whose role is to advise, advocate for, and coordinate all services the women Veteran may need [13].

Despite the expansion of women's health services within VHA, many women Veterans do not use VHA services [14]. Reasons for this are multi-faceted, including limited knowledge and inaccurate beliefs regarding eligibility for services [15] and their perspectives of their Veteran identity [16]. Accordingly, the VHA Office of Women's Health (OWH) developed the Women's Health Transition Training (WHTT) [17] pilot program in 2018 at five Air Force bases to inform women service members about their eligibility for VHA care and provide instructions on how to enroll to receive VHA care. The WHTT also seeks to increase women Veterans' awareness of VHA healthcare services for women (e.g., reproductive healthcare, maternity care, cancer screenings, military sexual trauma counseling) and change the misperception that VHA is only for men. In June 2019, WHTT became an official VHA program and expanded to the Army, Navy, and Marine Corps [18].

The WHTT has primarily been implemented with women transitioning out of military service. Yet specific groups of women Veterans, such as those residing in rural areas, may experience particularly salient barriers to using VHA services. Barriers for rural Veterans include the need to travel longer distances to receive care [19], fewer physicians, hospitals and health resources in rural areas, and limited broadband internet for telehealth [20]. These challenges can exacerbate health conditions [21, 22]. Thus, initiatives to ensure rural women Veterans' access to VHA care [22, 23] are essential.

Accordingly, programs that inform women Veterans about VHA service eligibility and comprehensive women's healthcare services may be particularly important for those living in rural areas. Nonetheless, knowledge of how to tailor such interventions (e.g., WHTT) for rural women Veterans not enrolled in VHA care is lacking. Additionally, although many studies have examined women Veterans' perceptions of VHA care, there is less knowledge of women

*not* enrolled in VHA care [16, 24, 25]. Moreover, despite increased focus on healthcare access and services for women Veterans and rural Veterans, there is limited knowledge of rural women Veterans' healthcare experiences [22, 23, 26, 27]. Understanding rural women Veterans' experiences and barriers to enrolling in VHA care is essential to tailoring interventions such as WHTT and successfully engaging more rural women Veterans in VHA healthcare to address their health needs.

In the current manuscript, we describe a VHA quality improvement focus group project conducted with rural women Veterans not enrolled in, or recently using, VHA care. Specifically, we sought to understand rural women Veterans' perceptions and needs regarding VHA healthcare, including barriers to enrolling in and using VHA care, and their perspectives on how to communicate with rural women Veterans about VHA services.

## Materials and methods

This project was a quality improvement project that was acknowledged as such by the local Department of Veteran Affairs (VA) Research and Development Committee, making it exempt by the local IRB.

### Eligibility

We aimed to conduct focus groups with rural women Veterans who had never enrolled in VHA care or who had not utilized VHA care in the past five years. Following insights from recent work [28] on rural identification, we permitted self-reporting of rural residence, rather than relying upon zip codes in the rural-urban commuting area (RUCA) dataset, to determine geographically rural designations; many women described their residence as rural even if not classified as such by their designated RUCA code [29]. Women had to be at least 18 years of age to participate.

### Recruitment

Recruitment efforts occurred both nationally and in specific locations where the team had established relationships, such as in Colorado, Washington, and Wisconsin. For national recruitment, we utilized Fiscal Year 2019 data from the United States Veterans Eligibility Trends and Statistics (USVETS), an integrated dataset of Veteran demographic data [30], to identify 2,500 rural women Veterans who were not currently enrolled in VHA care, stratified by region. In June and July 2021, we mailed letters and a flyer to these women Veterans to inform them of the focus group opportunity. We also disseminated a flyer to rural Veterans, rural community healthcare providers, and Veteran Service Officers (VSOs) through our team's community partnerships, Veteran-specific social media pages, and Veteran newsletters. Lastly, we attended women-specific monthly meetings of a Veterans Organization and staffed a booth at a women Veterans focused conference.

### Screening

Eighty-eight women Veterans contacted our team to express interest in participating in a focus group. Of those, 42.0% (n = 37) were eligible to participate upon telephone screening, during which a staff member obtained information from them regarding their demographics, military service, and enrollment and use of VHA care. 78.4% (n = 29) of those who were eligible during screening subsequently participated in a focus group. Common reasons for ineligibility included being unable to join the focus group virtually and no longer living in a rural location (Fig 1). Of focus group participants, the majority (n = 23; 79.3%) were recruited through USVETS mailings, while a smaller proportion were recruited through community engagement (n = 6; 20.7%).

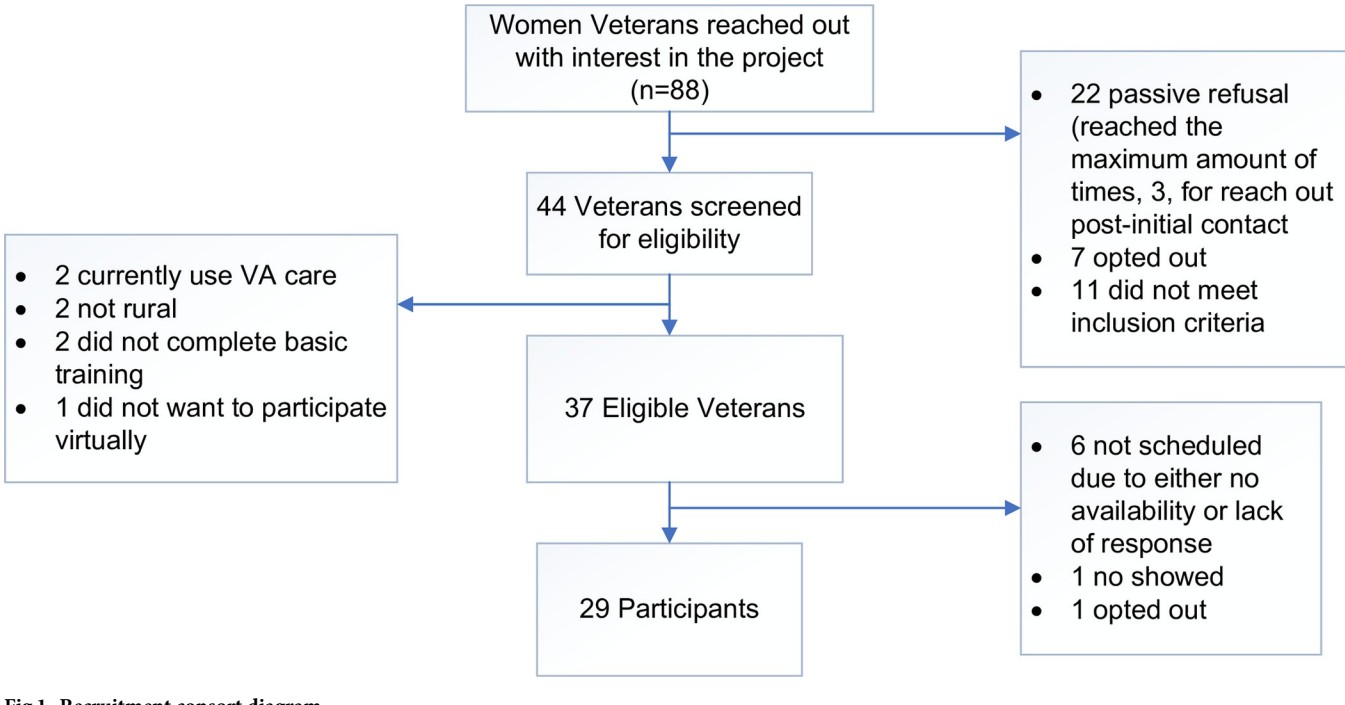

**Fig 1. Recruitment consort diagram.**

## Focus groups

In June and July 2021, we conducted 10 focus groups. Focus groups were selected because of their utility in investigating complex behavior and motivations about health behaviors [31], even when consensus among group members may be low [32]. Given COVID-19 constraints, focus groups were conducted virtually. Discussions were facilitated using a semi-structured guide (S1 File) by one of two facilitators. One facilitator was an established qualitative researcher; the other facilitator was a VA psychologist with content expertise regarding both women Veterans and rural Veterans, as well as experience qualitatively interviewing women Veterans. Information about the facilitators was provided to focus group participants. An additional team member took notes and managed technology. Focus groups were recorded for transcription and analysis, and informed verbal consent was obtained from focus group participants for taking part in the focus groups and recording of them. Recommended best practices for virtual data collection were followed [33, 34], including keeping focus groups small (2–4 Veterans per group) to facilitate deeper discussion. Each group lasted approximately 90 minutes. Veterans were emailed VHA resources (including information regarding resources for women Veterans) following each group.

## Analysis

Consistent with established qualitative methodology, analysis was conducted as a continuous process beginning with initial focus groups and continuing throughout and beyond the data generation period. Analysis occurred in an iterative and team-based process involving established qualitative content methods and reflexive team analysis [35, 36], led by an experienced qualitative methodologist (one of the focus group facilitators). Each transcript was read multiple times to achieve immersion prior to code development. The qualitative methodologist then

derived an initial code list deductively, based on domains included in the focus group guide, and inductively, based on additional domains emergent in transcripts. Following discussion and refinement by the project team, the final coding schema was applied to each transcript by the methodologist and analyzed together for patterns using the qualitative data software package ATLAS.ti [37, 38]. Throughout this process, team members met regularly to discuss emergent codes and themes and to assess preliminary results [39]. Thematic saturation was determined when repeated analysis yielded no new themes. Qualitative findings are reported according to the Consolidated Criteria for Reporting Qualitative Research (COREQ) 32-item qualitative standards checklist (S2 File) [40].

## Results

### Characterizing the sample: Demographics, military service, and VHA use

Participant characteristics are presented in Table 1. Participants tended to be older, and the majority were currently married. Racial and ethnic diversity was limited, with the majority identifying their race as White and none reporting Hispanic ethnicity. The majority were currently employed or retired. Branch of military service varied, with Army and Air Force most common. Most participants had not deployed. Time in service varied broadly. Most participants reported being honorably discharged. In terms of VHA enrollment and use, most participants have never enrolled in (or used) VHA care, while a small minority had previously enrolled in VHA care, though had not used any VHA services in the past five years. Focus group participants resided in the following states: Alaska (1), Washington (5), Oregon (1), Idaho (2), Montana (2) Wyoming (2), Utah (1), Colorado (2), Arizona (1), Nebraska (1), Iowa (1), Minnesota (2), Wisconsin (1), Illinois (2), Mississippi (1), Massachusetts (1), New Hampshire (1) and Maine (2).

### Overall focus group findings

Three primary themes emerged. First, participants reported poor communication about their eligibility for VHA services and the enrollment process. Second, they expressed the belief that VHA did not offer sufficient women's healthcare services. Finally, they perceived the inconvenience of accessing VHA facilities as a barrier to using VHA healthcare.

**Poor communication about eligibility and enrollment.** An overwhelming majority of participants indicated a lack of communication and/or miscommunication regarding their eligibility for VHA healthcare and the process of determining if they were eligible to obtain VHA services. Indeed, not a single participant described perceiving that there had been clear communication about eligibility. Instead, confusion about this was typically cited as the primary barrier to enrollment in VHA care. Veterans described their strong perception that, in their own experience and the experiences of other women Veterans whom they had interacted with, information about what VHA services they qualified for and how to access services (i.e., process, location) was not disseminated effectively across VA or military systems. This resulted in the perception of opaque and/or inequitable eligibility.

*I haven't filled out the [VA healthcare] paperwork just cause, you know, it's paperwork. . .and then you get a rejection or no, you're not eligible, and you've put five hours into filling out. . . 50 pages of whatever, and it just seems like bureaucracy again. It's the whole process by which you. . . why not just make it automatic, why not just tell everybody, hey, you're, you're in the VA system. Just go and show up. Why do you have to fill out all this paperwork to prove that you're a person that got out of the military?. . . In this day and age with computers and the fact that you [project team] found us, why isn't the VA finding us?*

**Table 1. Focus group participant characteristics (n = 29).**

| Characteristic | M | SD | n (%) |
|---|---|---|---|
| **Age** (years) | 60 | ± 10 | - |
| **Military Service** (years) | 7 | ± 6 | - |
| **Marital Status** | - | - | |
| Married | | | 23 (79%) |
| Single and never married | | | 4 (14%) |
| Divorced | | | 0 (0%) |
| Widowed | | | 2 (7%) |
| **Hispanic/Latina** | - | - | |
| Yes | | | 0 (0%) |
| No | | | 29 (100%) |
| **Race** | - | - | |
| American Indian or Alaska Native | | | 1 (3%) |
| Black or African American | | | 2 (7%) |
| White | | | 26 (90%) |
| **Employment** | - | - | |
| Employed full-time | | | 12 (41%) |
| Employed part-time | | | 4 (14%) |
| Not employed, not currently seeking assistance | | | 2 (7%) |
| Retired from the workforce | | | 11 (38%) |
| **Military Branch** | - | - | |
| Air Force | | | 9 (31%) |
| Army or Army National | | | 16 (55%) |
| Guard | | | 1 (3%) |
| Marine Corps | | | 3 (11%) |
| Navy | | | |
| **Ever Deployed** | - | - | |
| Yes | | | 8 (28%) |
| No | | | 21 (72%) |
| **Discharge Status** | - | - | |
| Honorable | | | 27 (93%) |
| General (Medical) | | | 1 (3%) |
| Other than Honorable | | | 1 (3%) |
| **VHA Enrollment and Use** | - | - | |
| Never enrolled, no use | | | 27 (93%) |
| No use in the last five years | | | 2 (7%) |

To improve understanding of VHA eligibility and streamline the process for determining and/or obtaining services, participants emphasized the need for widespread marketing and consistent messaging as well as, in some cases, more individualized communication and outreach to women Veterans to inform them of services available, eligibility nuances, and VHA points of contact for interfacing with the system. Participants suggested immediately preceding military discharge as one critical time to effectively communicate such information, since transitioning service members are a receptive population and have not yet disbursed, taken new employment, and/or obtained other health insurance. Participants noted that it is precisely then that women most need to understand what they will qualify for after discharge and what exactly they must do to access care.

*Facilitator: So, it sounds like there's a theme that's emerging here, that there's some miscommunication or lack of communication about eligibility, about process, just about all the details.*

*Participant 1: Yeah. That actually could be helpful when you get out [of the military at discharge] . . .if somebody wants benefits or understanding how they could get them.*

*Participant 2: It would've been nice. . .when I got discharged if somebody would've said, hey. . . don't forget, you're eligible for this. Here's how you look into it. . . give you some little heads-up, but I got nothing.*

Because of the clarity needed for this information, Veterans also suggested that better training about eligibility and enrollment should be required for branch officers tasked with discharge, since those points of contact should be able to answer basic questions and provide initial guidance about next steps for enrolling in VHA healthcare.

*Participant 1: When I separated, they didn't say what, what the benefits were. . .what your eligibility was.*

*Participant 2: I'm not sure if it was more the VA or if it was more of the military itself, the Air Force itself. Because, I mean, they're the ones that gave me my discharge papers. . . And they're the ones that gave me the impression that I was not eligible for anything unless it happened while I was in the military.*

*Participant 3: I was gonna say the exact same thing. I'm not sure they really knew. They should know so they can tell people.*

Military separation was not the only time noted, however, as important for obtaining information regarding VHA eligibility and enrollment. Women also expressed a desire to receive information about their eligibility for VHA services around key transitions in their lives, such as retirement from subsequent (non-military) employment, as this often represented a period of transition with substantial impacts to their healthcare coverage.

*Participant: Yeah, remind them, and tell them that there is a way to find out more information about it if they're interested. I know maybe had I gotten that ten years ago or something, I might have. . . checked it out more, you know, because health does change when you get older. . . When you're in your 30s and 40s, you may not think about it so much if you're. . . out by then, but. . . when you hit your 70s, things start escalating and changing, so I think that routinely, letters should be sent out so that Veterans do know what is available.*

Women also emphasized the need for the VA to reach out to women Veterans through multiple modalities, with clear information about eligibility and the process of determining it. Various methods for sharing information about VHA services offered, eligibility, and enrollment processes were suggested. Most emphasized was the need to improve and simplify the VA website so the process of determining VHA eligibility is clear, including providing contact numbers or email addresses of people whom Veterans can communicate with concerning personal questions.

*Participant 1: I don't think you can go to an online thing and find out automatically, quickly, if you are eligible for services. That would've been awesome, and I think that that would be*

*true for a lot of people if there was a quick way of finding out that you're eligible, like just on the website, then I think more people would access it.*

*Participant 2: That's what we need. End the uncertainty, just making it quick, easy. You go here, and then you get an answer right away. Or even just a name and number, a real person we can direct contact.*

Other suggested methods included direct mailings to Veterans, direct outreach to Veteran Service Organizations and/or local care facilities and other rural community resources, and advertising in communities, Veteran newsletters, and television.

*Participant 1: I think there's two times [when it would be helpful to receive information about VA healthcare]. Number one is when someone is discharged, for them to get a whole packet giving them all the information about the VA and their options. And then. . . maybe at some point, there should be a general letter sent out to Veterans. . . maybe every ten years or what-ever, they send out a big blanket folder and things to let people know about the services that they could get through the VA and tell them that please apply, and you'll get further informa-tion [about eligibility].*

**Perception of lack of women-focused care.** Many focus group participants reported the perception that VHA does not provide healthcare oriented to women. Some cited this as disin-centive to enrolling in or using VHA healthcare.

*Participant 1: They never show you any women Veterans in those news stories [about VA healthcare]. They're always men. So women Veterans, they're a very well-kept secret because we know nothing about them.*

*Participant 2: I agree 100%, 100%.*

*Participant 3: I agree as well. And out of all the women Veterans I know, I don't know any of them that go to the VA. And I've been in four or five different states with my careers, and it's all these Veterans, they don't go to the VA, and so even the ones that have retired don't use the VA.*

Some participants explicitly associated VHA with Veteran men, particularly those of older generations.

*Facilitator: What are your impressions or your thoughts when you think about VA health care?*

*Participant 1: Old white men.*

*Facilitator: Old white men. OK. You mean as the doctor, or as the provider, or?*

*Participant 1: No, as the patients, and you know, when I get around a bunch of Veterans, like VFW Veterans or Legion Vets, it gives me flashbacks to when I was in the military and had to put up with sexism and stuff like that, so it doesn't really jive for me.*

*Participant 2: I think, even when I was active duty and working, the thought of having to enter the VA system was honestly a bit sketchy to me and scary because it was, you know, a bunch of old men dying in hospitals is what it felt like to me. . .*

Some participants cited concerns about the quality of women's healthcare services within VHA, as well as regarding insensitivity to women's privacy needs.

*Participant 1: I wouldn't go to the VA if I had breast cancer... I wouldn't trust the VA with that. I would probably go more with a specialist or somebody who is more in line with that type of care... If I had actually known that there was somebody right down the street that I could go see, I would probably only see the VA mostly for check-ups.*

*Facilitator: And is that because, is it a specialization issue, like if there was a specialist at the VA in that, that would be fine? Or is it a concern about possible quality of care?*

*Participant 1: If I found a specialist that leaned that direction, I would probably feel a little more comfort for it, but I would venture to say it's mostly on the horror stories of quality of care and past experience that I would most likely go a different direction.*

*Participant 2: The year I was [in Veteran Service Organization, we had town hall meetings, and we were told by VA reps that the only thing that was available for female Veterans for their yearly breast and gyn exams were telehealth, and I said you expect me to go out and tell my female Veterans that they're going to have to be on a television screen for their breast or gyn exam, and the answer was yes... so I think they're insensitive to it as well.*

For the aforementioned reasons, many participants expressed strong interest in women's specialty clinics. Most focus group participants were not aware that women's specialty clinics exist within VHA facilities. Some also indicated support for having more female physicians in VHA.

*Participant 1: I think that [getting care at a VA women's clinic] would be lots more comfortable for all of us, well, it certainly would be for me, but, you know, with the current system, I don't think that's out there anywhere, is it?*

*Participant 2: With more women physicians.*

*Participant 3: I don't care if I have a male or a woman physician, as long as they're there to do their job, and they do their job well, looking after me the way I need to be looked after and not making me feel like a number. But a women-only location, I think that that would actually be an amazing idea because then they're specialized in that one field. I mean, we are completely different than men... I would say that yeah, a woman-only clinic would be an amazing idea.*

Participants stressed that, to increase women Veterans' use of VHA services, any outreach and/or advertising should explicitly present the VHA as a place of care for women, to counteract the association of VHA as a bastion of "old White men." This would include advertisement of women's clinics and services available within VHA.

**Inconvenience of accessing services.** A third and final barrier centered on the inconvenience of accessing VHA services. Some participants noted that many VHA facilities were geographically distant from the rural communities in which they lived and thus took significant time to drive to.

*I was told that I had to have my initial evaluation by a civilian provider that was on the [direction] side of [city], which is a four-hour drive from my home, and I had an 8:00 in the morning appointment, and so I asked could I make arrangements to go the evening before and be there, and I was told, well, I could, but I'd have to foot the bill, there were nothing to help compensate for it, and I said well, there's a VA clinic in [a smaller city], which is 65 miles*

*away and then there's one in [city] which is about 75 miles away, and I said I could go to either of those, and I could drive that relatively easy and be there by 8:00 in the morning, and they said no, that's not where you'll go. You have to go to this place on the [direction] side of [city], and I said, I'm not going there, and they said, OK, we're going to put down on your application that you refused the appointment that we gave you.*

Some focus group participants also reported perceived difficulty getting timely appointments as a barrier to using VHA care.

*For me, it's a lack of care. I know lots and lots of people who are within the VA, but they still can't get an appointment. You had a high blood pressure, you have a nosebleed, and they say no, you're not authorized to go to the emergency room. You've got to wait and come to our clinic on Monday when you can ask for an appointment, and the appointment is six weeks down the road. So, for me, it's the inability of availability.*

Further, some participants reported that technological limitations in their rural areas made virtual appointments difficult. For example, some had difficulty accessing broadband internet from home, and expressed the desire for other (e.g., mobile) options for in-person care:

*Participant: It's a two-hour drive to go to a VA center, so not that small towns necessarily need a permanent VA structure, but they should have that mobile bus. . .. [the public health department] will come out here and do mobile mammograms, and the VA should do the same thing. They should have a mobile office that hits these small-town areas because it is a burden for somebody to have to drive two hours to go to the doctor.*

*Facilitator: If the VA offered any services to women Veterans in rural areas that could be delivered virtually, is that something that would be of interest or not?*

*Participant: At home, I have satellite internet. Virtual doctor visits and things like that don't work well. I happen to be in town for this [virtual focus group] right now, so that way I can. . .basically [use] somebody else's internet for this. . ..because it would use all my data at home.*

Several others indicated that, even if VHA telehealth services were possible, they would not want to use telehealth services, as telehealth was not perceived as appropriate (e.g., OB/GYN) or personal enough (e.g., mental health) for their healthcare needs.

Because of the perceived inconveniences of accessing VHA services, some focus group participants who had other health insurance options preferred to receive care at a closer non-VHA facility. Participants also noted that many Veterans receive healthcare through both private healthcare organizations and VHA, and that better communication and coordination between them is necessary to reduce record scatter (different records in multiple systems), focus resources, and improve patient service and experiences. Similarly, some emphasized the need to build and/or strengthen connections to and partnerships with local community care facilities to improve healthcare access. These participants described a desire for a system in which they could show up to any healthcare facility with a Veteran card and receive healthcare there. However, a few noted concerns about privacy within their rural communities and noted that, as such, they would like to be able to access VHA healthcare, if eligible.

## Discussion

Perceptions about poor communication about eligibility and enrollment, insufficient women's healthcare services, and inconvenience of accessing VHA facilities were described as barriers

to enrolling in and using VHA care among rural women Veterans who participated in our focus groups.

## Improving communication

Our findings suggest that lack of accurate information regarding eligibility for VHA healthcare and enrollment, as well as regarding healthcare services available to women Veterans, precluded rural women Veterans from enrolling in and using VHA services. Additionally, consistent with prior studies of women Veterans who utilized VHA care [41, 42], rural, unenrolled women Veterans believed that VHA does not provide healthcare relevant to women Veterans' unique needs and that VHA is a place for "old White men," despite significant VHA expansion of healthcare services for women Veterans to include comprehensive women's health services, maternity care coordination, and provision of specialized training to VHA Women's Health providers in women Veterans' healthcare needs [43–46]. As many women Veterans have experienced their needs as being disregarded or have experienced prior interpersonal or institutional trauma [47–49], these perceptions may be particularly detrimental, as negative perceptions regarding VHA care can result in delayed or forgone healthcare use [50, 51].

Accordingly, it is critical to ensure that accurate information is widely available to rural women Veterans regarding VHA eligibility, the process of VHA enrollment, and women's health services. Such information should be easily accessible, consistently updated, and disseminated in simple, clear terms through multiple modalities. Updates should be disseminated to rural women Veterans, irrespective of when they served, and to those who interact with rural women Veterans. For example, VSOs, community healthcare providers, and branch officers involved in military separation may be important conduits of such knowledge. Local VHA facilities could also periodically reach out to rural women Veterans through targeted mailings or community events to provide information regarding services available to women Veterans, how to access those services, as well as specific ways in which VHA is responding to women Veterans' healthcare needs and concerns. For example, VHA has implemented broad initiatives to prevent harassment at VHA facilities and ensure its facilities are welcoming to women Veterans. As rural women Veterans in our sample were generally unaware of these efforts, such information could be more widely disseminated outside of VHA.

Efforts to communicate such information with rural women Veterans should consider the timing of such communications. Although some women Veterans in our sample reported that they did not need to use VHA services in the period following separation from the military, a portion described later needing healthcare services, indicating that they would have used or considered using VHA care at other points in their lives, if they were eligible to use VHA care. However, many were unsure if they were eligible or did not know how to enroll in VHA care, and women were largely unaware that eligibility can change over time or be offered for specific circumstances or types of care (e.g., healthcare for conditions related to military sexual trauma; time-limited mental healthcare). Thus, targeted communications with rural women Veterans not enrolled in or using VHA services could emphasize both the period preceding military separation, as well as other life transitions where healthcare services may be particularly important (e.g., employment-related transitions, such as retirement). Additionally, such communications could clarify exceptions to broader eligibility to inform women Veterans of exceptions to eligibility for VHA care and circumstances in which their eligibility could change.

## Inconvenience of accessing VHA services

Lack of access and transportation, which have been noted to also impede use of healthcare services for rural Veterans in other studies [52–54], were noted barriers among rural women

Veterans to using VHA services in the present sample. Thus, it is critical to determine feasible, acceptable, and effective ways to deliver healthcare to rural women Veterans. Telehealth may be a particularly important modality of healthcare among rural women Veterans, particularly considering barriers within this population (e.g., caregiving) which may further compound access-related challenges [55]. As access to reliable broadband internet in rural areas can impede access to telehealth, it may be important to establish hubs in rural communities where Veterans can privately access internet for telehealth. Additionally, it may be particularly important to increase rural women Veterans' awareness of the VA Digital Divide program, which offers discounts on home internet and phone services or provides a device with internet connectivity for Veterans to engage in telehealth [56]. While VHA offers some such services at community-based outpatient clinics (CBOCs), it may be helpful to also establish these in easily accessible locations which do not require extensive transportation to access. Communication between VHA and community healthcare organizations is critical to ensure service coordination and provision. Finally, in communities that offer transportation for rural Veterans (e.g., carpooling to VHA facilities), it is important to ensure that women Veterans, VSOs, and community providers are all aware of these offerings.

## Limitations

One major limitation to our findings is that some women whom we interviewed reported that they were not eligible for VHA care, yet we were unable to verify this and did not ask women to elaborate on the last time they had applied for VHA care. Eligibility has expanded widely, and thus many of these women may now be eligible. Findings are also limited by the focus on specific geographic areas and the lack of racial and ethnic diversity within our sample. Additional research is needed to understand how findings extend to rural women Veterans of other racial (e.g., Black, Asian American, Pacific Islander, American Indian, Native Alaskan) and ethnic (i.e., Hispanic) backgrounds, and to understand the perceptions and experiences of rural women Veterans in other geographic areas where there may be additional concerns and barriers to enrolling in VHA care (e.g., US Territories).

## Conclusion

Rural women Veteran focus group participants reported several barriers to enrollment and use of VHA care, including poor communication regarding VHA eligibility and enrollment, negative perceptions of VHA services for women, and inconvenient access to VHA facilities. These barriers can be addressed by improving communication about VHA eligibility and enrollment, increasing awareness regarding VHA healthcare services for women Veterans, and addressing barriers related to access and transportation in rural locations. These findings and recommendations can be used to inform modifications to the WHTT intervention, as well as development of improved marketing and dissemination of information to rural, unenrolled women Veterans to ensure they can access VHA healthcare when needed.

## Supporting information

**S1 File. Focus group interview guide.**
(PDF)

**S2 File. Consolidated criteria for reporting qualitative research.**
(PDF)

## Acknowledgments

The authors would like to gratefully acknowledge each of the focus group participants for sharing their time and perspectives. The authors would also like to thank our collaborators from the Denver COIN, Rocky Mountain MIRECC, VA Office of Rural Health (ORH) Growing Rural Outreach through Veteran Engagement (GROVE) Center, and ORH VRHRC-Portland.

## Author Contributions

**Conceptualization:** Carly M. Rohs, Karen R. Albright, Lindsey L. Monteith, Amber D. Lane, Kelty B. Fehling.

**Data curation:** Carly M. Rohs, Karen R. Albright, Lindsey L. Monteith, Kelty B. Fehling.

**Formal analysis:** Karen R. Albright.

**Funding acquisition:** Kelty B. Fehling.

**Investigation:** Carly M. Rohs, Lindsey L. Monteith, Kelty B. Fehling.

**Methodology:** Carly M. Rohs, Karen R. Albright, Lindsey L. Monteith, Amber D. Lane, Kelty B. Fehling.

**Project administration:** Carly M. Rohs, Amber D. Lane, Kelty B. Fehling.

**Resources:** Lindsey L. Monteith, Amber D. Lane, Kelty B. Fehling.

**Software:** Carly M. Rohs, Amber D. Lane, Kelty B. Fehling.

**Supervision:** Karen R. Albright, Lindsey L. Monteith, Kelty B. Fehling.

**Validation:** Karen R. Albright, Lindsey L. Monteith, Kelty B. Fehling.

**Visualization:** Carly M. Rohs, Karen R. Albright, Lindsey L. Monteith, Amber D. Lane, Kelty B. Fehling.

**Writing – original draft:** Carly M. Rohs, Kelty B. Fehling.

**Writing – review & editing:** Carly M. Rohs, Karen R. Albright, Lindsey L. Monteith, Amber D. Lane, Kelty B. Fehling.

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
