## [Decision Letter · Decision Letter 0]

14 Apr 2023

PONE-D-23-00940Perspectives of VA Healthcare from Rural Women Veterans not Enrolled in or using VA HealthcarePLOS ONE

Dear Dr. Rohs,

Thank you for submitting your manuscript to PLOS ONE. My apologies that it has taken some time to get back to you regarding your manuscript as it was incredibly difficult to secure appropriate reviewers. After careful consideration, we feel that it has merit but does not fully meet PLOS ONE’s publication criteria as it currently stands. Therefore, we invite you to submit a revised version of the manuscript that addresses the points raised during the review process.

We look forward to receiving your revised manuscript.

Kind regards,

Mathew Albert Wei Ting Lim

Academic Editor

PLOS ONE

Journal Requirements:

3. We note that Figure 1 in your submission contain map images which may be copyrighted. All PLOS content is published under the Creative Commons Attribution License (CC BY 4.0), which means that the manuscript, images, and Supporting Information files will be freely available online, and any third party is permitted to access, download, copy, distribute, and use these materials in any way, even commercially, with proper attribution. For these reasons, we cannot publish previously copyrighted maps or satellite images created using proprietary data, such as Google software (Google Maps, Street View, and Earth). For more information, see our copyright guidelines: http://journals.plos.org/plosone/s/licenses-and-copyright.

Reviewers' comments:

Reviewer's Responses to Questions

**Comments to the Author**

1. Is the manuscript technically sound, and do the data support the conclusions?

Reviewer #1: Partly

Reviewer #2: Yes

2. Has the statistical analysis been performed appropriately and rigorously? 

Reviewer #1: N/A

Reviewer #2: N/A

3. Have the authors made all data underlying the findings in their manuscript fully available?

Reviewer #1: No

Reviewer #2: No

4. Is the manuscript presented in an intelligible fashion and written in standard English?

Reviewer #1: Yes

Reviewer #2: Yes

5. Review Comments to the Author

Reviewer #1: The authors describe findings from qualitative interviews with rural-residing women Veterans who are not enrolled in or have not recently used VHA care. Given the focus of the VHA on improving access to and quality of care for women and rural Veterans, this is an important topic. The manuscript could be strengthened by addressing the following:

Abstract

• Include information about the analytical approach

Introduction

• VA-specific information about the women veterans program managers and other efforts will likely not be understood by a non-VA audience. Some grounding in the ways in which VA has attempted to improve care access and quality is needed.

• It’s unclear if the WHTT targets rural women specifically. The framing initially seems that way.

• In some places the writing could be streamlined, and ideas/sentences combined for better flow.

• The rationale for the study could be strengthened. In particular, what might be unique about rural veterans that necessitates a study specifically in this population?

Methods

• The order of the subheadings could be improved. For example, the consent and recording information should not be ahead of the study population and recruitment methods.

• The authors note the “narrow recruitment timeframe” as a limitation to how they could identify participants. It is not clear why there was a narrow timeframe.

• Additional information should be included regarding local dissemination efforts. How was the flyer disseminated to rural women and what is “local”?

• For the use of USVETS, did you specifically target women without VHA insurance? As written, it almost suggests targeting of women who do have VHA insurance.

• The description of analytic approach is weak and should be bolstered. For example, who was involved in coding and how, who was involved in identifying themes, and what is the positionality/background of those involved in these elements? The content analysis approach should also be described in more detail. How did you determine if thematic saturation or information power was reached?

Results

• The quotes in the results are strong. Were there any outliers to the themes? For example, participants who knew about VA and their eligibility but chose not to use it? Any who perceived VA had good quality care but didn’t use it for one reason or another?

Discussion

• The discussion would be more compelling if the findings were integrated and discussed together rather than laid out in separate paragraphs. For example, theme 2 reflected women lacking important information (theme 1) that might change their perceptions (not knowing that there are women’s clinics or designated women’s health providers).

• How do your findings relate to any previous literature on lack of information on eligibility and services?

• Is there any overlap in findings with qualitative literature among VA users?

• Whether and how the findings differ for rural women vs. all women Veterans isn’t clear. Are there specific themes or elements of the themes that are unique to rural women Veterans (broadband internet, which is mentioned, seems to be one). Any others?

• Regarding technology barriers or lack of broadband internet, you may want to consider discussing VA’s digital divide consult, which is intended to help improve access to broadband, as well as mobile devices.

Reviewer #2: The article thoroughly addresses all aspects of the COREQ checklist for qualitative research. It follows a logical progression and sound reasoning. Findings support conclusions.

Line #235: Someone’s name is included in the quote. I believe that should be stricken out with a parenthetical note substituted to keep everything anonymous.

6. PLOS authors have the option to publish the peer review history of their article (what does this mean?). If published, this will include your full peer review and any attached files.

Reviewer #1: No

Reviewer #2: No

---

## [Author Response · Author response to Decision Letter 0]

13 Jun 2023

Thank you for taking the time to review our manuscript. I have included responses in the attached "Response to Reviewers" document and below. 

We would like to thank the reviewers and PLOS ONE editors for their helpful feedback and comments. When referring to line numbers below, these reflect line numbers in the track changes version of the manuscript.

Academic Editor

1. Please ensure that your manuscript meets PLOS ONE's style requirements, including those

for file naming. The PLOS ONE style templates can be found at

RESPONSE:

Thank you for providing this information. We have reviewed this information to ensure that the revised files meet these requirements. 

2. In your Data Availability statement, you have not specified where the minimal data set

underlying the results described in your manuscript can be found. PLOS defines a study's

minimal data set as the underlying data used to reach the conclusions drawn in the manuscript

and any additional data required to replicate the reported study findings in their entirety. All

PLOS journals require that the minimal data set be made fully available. For more information

about our data policy, please see http://journals.plos.org/plosone/s/data-availability.

Upon re-submitting your revised manuscript, please upload your study’s minimal underlying

data set as either Supporting Information files or to a stable, public repository and include the

relevant URLs, DOIs, or accession numbers within your revised cover letter. For a list of

acceptable repositories, please see http://journals.plos.org/plosone/s/data-availability#locrecommended-

repositories. Any potentially identifying patient information must be fully

anonymized.

Important: If there are ethical or legal restrictions to sharing your data publicly, please explain

these restrictions in detail. Please see our guidelines for more information on what we consider

unacceptable restrictions to publicly sharing data : http://journals.plos.org/plosone/s/dataavailability# loc-unacceptable-data-access-restrictions. 

Note that it is not acceptable for the authors to be the sole named individuals responsible for ensuring data access. We will update your Data Availability statement to reflect the information you provide in your cover letter.

RESPONSE: 

Institutional restrictions prohibit us from sharing this data publicly. Specifically, we do not have approval by our regulatory authority (the VA Rocky Mountain Regional VA Research and Development Committee) to share de-identified data publicly for this study. Rather, de-identified data can be accessed with a Data Use Agreement and verification of IRB approval from the requestor. We have provided contact information for our local regulatory authority (the VA Rocky Mountain Regional VA Research & Development Committee) to review any such requests.

3. We note that Figure 1 in your submission contain map images which may be copyrighted.

All PLOS content is published under the Creative Commons Attribution License (CC BY 4.0),

which means that the manuscript, images, and Supporting Information files will be freely

available online, and any third party is permitted to access, download, copy, distribute, and use

these materials in any way, even commercially, with proper attribution. For these reasons, we

cannot publish previously copyrighted maps or satellite images created using proprietary data,

such as Google software (Google Maps, Street View, and Earth). For more information, see

our copyright guidelines: http://journals.plos.org/plosone/s/licenses-and-copyright.

We require you to either (1) present written permission from the copyright holder to publish

these figures specifically under the CC BY 4.0 license, or (2) remove the figures from your

submission:

a. You may seek permission from the original copyright holder of Figure 1 to publish the

content specifically under the CC BY 4.0 license. 

We recommend that you contact the original copyright holder with the Content Permission Form (http://journals.plos.org/plosone/s/file?id=7c09/content-permission-form.pdf) and the following text: “I request permission for the open-access journal PLOS ONE to publish XXX under the

Creative Commons Attribution License (CCAL) CC BY 4.0 (http://creativecommons.org/licenses/by/4.0/). Please be aware that this license allows

unrestricted use and distribution, even commercially, by third parties. Please reply and provide

explicit written permission to publish XXX under a CC BY license and complete the attached

form.”

Please upload the completed Content Permission Form or other proof of granted permissions

as an "Other" file with your submission.

In the figure caption of the copyrighted figure, please include the following text: “Reprinted

from [ref] under a CC BY license, with permission from [name of publisher], original

copyright [original copyright year].”

b. If you are unable to obtain permission from the original copyright holder to publish these

figures under the CC BY 4.0 license or if the copyright holder’s requirements are incompatible

with the CC BY 4.0 license, please either i) remove the figure or ii) supply a replacement

figure that complies with the CC BY 4.0 license. Please check copyright information on all

replacement figures and update the figure caption with source information. If applicable,

please specify in the figure caption text when a figure is similar but not identical to the

original image and is therefore for illustrative purposes only.

The Gateway to Astronaut Photography of Earth (public domain):

http://eol.jsc.nasa.gov/sseop/clickmap/

Maps at the CIA (public domain): https://www.cia.gov/library/publications/the-worldfactbook/

index.html and https://www.cia.gov/library/publications/cia-mapspublications/

index.html

USGS EROS (Earth Resources Observatory and Science (EROS) Center) (public domain):

http://eros.usgs.gov/#

RESPONSE:

1. Thank you for noting this. We have decided to remove this figure from the manuscript and included the statement in lines 215-218: “Focus group participants resided in the following states: Alaska (1), Washington (5), Oregon (1), Idaho (2), Montana (2) Wyoming (2), Utah (1), Colorado (2), Arizona (1), Nebraska (1), Iowa (1), Minnesota (2), Wisconsin (1), Illinois (2), Mississippi (1), Massachusetts (1), New Hampshire (1) and Maine (2).”

Reviewer #1: 

Abstract

Include information about the analytical approach

Response: We have added this information to the Abstract and now state: “A thematic inductive analytic approach was used to analyze focus group transcripts (line 42).”

Introduction

VA-specific information about the women veteran’s program managers and other efforts will

likely not be understood by a non-VA audience. Some grounding in the ways in which VA has

attempted to improve care access and quality is needed.

Response: We agree and have added language regarding healthcare for women Veteran in the Introduction. We now state: “As such, it is essential that women Veterans have access to the comprehensive healthcare services provided by the Veterans Health Administration (VHA), which has made concerted efforts to address women Veterans’ specific healthcare needs through women’s health clinics and Patient Aligned Care Teams that work closely with Women Veterans Program Managers in the provision of reproductive healthcare and maternity care coordination (10-12). The VHA has committed to having a WVPM located in every regional office, whose role is to advise, advocate for, and coordinate all services the women Veteran may need (13) (lines 89-95).”

It’s unclear if the WHTT targets rural women specifically. The framing initially seems that way.

Response: The WHTT was not initially developed to target rural women Veterans specifically. We have clarified this and now state: “Accordingly, the VHA Office of Women’s Health (OWH) developed the Women’s Health Transition Training (WHTT) (17) pilot program in 2018 at five Air Force bases to inform all women service members about their eligibility for VHA care and provide instructions on how to enroll to receive VHA care. In June 2019 it became an official VHA program and expanded to Army, Navy, and Marine Corps (18). The WHTT also seeks to increase women Veterans’ awareness of VHA healthcare services for women (e.g., reproductive healthcare, maternity care, cancer screenings, military sexual trauma counseling) and change the misperception that VHA is only for men (lines 99-105).”

In some places the writing could be streamlined, and ideas/sentences combined for better

flow.

Response: Thank you for this comment. We have revised the writing throughout the manuscript to streamline ideas and improve flow. 

The rationale for the study could be strengthened. In particular, what might be unique about

rural veterans that necessitates a study specifically in this population?

Response: We have included added content to bolster the rationale for focusing on women Veterans, including adding literature that further underscores the additional barriers that rural Veterans experience in accessing VHA healthcare. 

We included content throughout the introduction which a portion now reads: 

“The WHTT has primarily been implemented with women transitioning out of military service. Yet specific groups of women Veterans, such as those residing in rural areas, may experience particularly salient barriers to using VHA services. Barriers for rural Veterans include the need to travel longer distances to receive care (19), fewer physicians, hospitals and health resources in rural areas, and limited broadband internet for telehealth (19). These challenges exacerbate health conditions (21,22). Thus, initiatives to ensure rural women Veterans’ access to VHA care (22,23) are essential. 

 Accordingly, programs that inform women Veterans about VHA service eligibility and comprehensive women’s healthcare services may be particularly important for those in rural areas. Nonetheless, knowledge of how to tailor such interventions (e.g., WHTT) for rural women Veterans not enrolled in VHA care is lacking. Additionally, although many studies have examined women Veterans’ perceptions of VHA care, there is less knowledge of women not enrolled in VHA care (16,24,25). Moreover, despite increased focus on healthcare access and services for women Veterans and rural Veterans, there is limited knowledge of rural women Veterans’ healthcare experiences (22,23,26,27). Understanding rural women Veterans’ experiences and barriers to enrolling in VHA care is essential to tailoring interventions such as WHTT and successfully engaging more rural women Veterans in VHA healthcare to address their health needs (lines 106-121).” 

Methods

The order of the subheadings could be improved. For example, the consent and recording

information should not be ahead of the study population and recruitment methods.

Response: As suggested, we have moved content to fit more appropriately into subheadings. For example, consent and recording information are now instead included in lines 181-182 under the “Focus Groups” subheading, after the “Eligibility,” “Recruitment,” and “Screening” subheadings. With such revisions, altering the order of subheadings no longer seemed warranted. 

The authors note the “narrow recruitment timeframe” as a limitation to how they could

identify participants. It is not clear why there was a narrow timeframe.

Response: We apologize for this error, which was misstated previously; our inclusion criteria were not changed due to our recruitment timeframe. We have rectified this and now state: “We aimed to conduct focus groups with rural women Veterans who had never enrolled in VHA care or had not utilized VHA care in the past five years (lines 137-140).”

Additional information should be included regarding local dissemination efforts. How was

the flyer disseminated to rural women and what is “local”?

Response: 

We have added expanded our description of recruitment, noting that the mailing included a letter, plus the flyer, and describing the methods used to disseminate the flyer through mailings, community partnerships, and social media pages. 

In lines 146-150, we clarified that our concentrated efforts focused on the states in which the team had established relationships, “Recruitment efforts occurred both nationally and in specific locations in which the team had established relationships with caregivers, such as in Colorado, Washington, and Wisconsin.”

We also added in lines 154-158: “In June and July 2021, we mailed letters and a study flyer, to these women Veterans to inform them of the focus group opportunity. We also disseminated a flyer to rural Veterans, rural community healthcare providers, and Veteran Service Officers (VSOs) through our team’s community partnerships, Veteran-specific social media pages, and Veteran newsletters.”

Lastly, we removed the term “local” from the manuscript to remove confusion that recruitment only focused locally. 

For the use of USVETS, did you specifically target women without VHA insurance? As

written, it almost suggests targeting of women who do have VHA insurance.

Response: We targeted women without VHA insurance and have clarified the language. In lines 150-153, we state: “For national recruitment, we utilized Fiscal Year 2019 data from the United States Veterans Eligibility Trends and Statistics (USVETS), an integrated dataset of Veteran demographic data (30), to identify 2,500 rural women Veterans who were not currently enrolled in VHA care, stratified by region.”

The description of analytic approach is weak and should be bolstered. For example, who was involved in coding and how, who was involved in identifying themes, and what is the

positionality/background of those involved in these elements? The content analysis approach should also be described in more detail. How did you determine if thematic saturation or

information power was reached?

Response: All these important questions have now been edited in the analytic section, including specification of who was involved in coding and how, who identified themes and how, and the background of the individual involved. We have added clarification on the content analysis approach and regarding how thematic saturation was reached. All edited text can be found in lines 193-202 and is included below: “The qualitative methodologist then derived an initial code list deductively, based on domains included in the focus group guide, and inductively, based on additional domains emergent in transcripts. Following discussion and refinement by the study team, the final coding schema was applied to each transcript by the methodologist and analyzed together for patterns using the qualitative data software package ATLAS.ti (37,38). Throughout this process, team members met regularly to discuss emergent codes and themes and to assess preliminary results (39). Thematic saturation was determined when repeated analysis yielded no new themes. Qualitative findings are reported according to the Consolidated Criteria for Reporting Qualitative Research (COREQ) 32-item qualitative standards checklist (S2 File) (40).”

Results

The quotes in the results are strong. Were there any outliers to the themes? For example, participants who knew about VA and their eligibility but chose not to use it? Any who perceived VA had good quality care but didn’t use it for one reason or another?

Response: Thank you. We have carefully re-reviewed the Results with this question in mind. For the theme “Poor communication about eligibility and enrollment.” We added the following: “Indeed, not a single participant described clear communication about eligibility (lines 265-266).”

For the themes, “Perception of lack of women-focused care,” and “Inconvenience of accessing services,” we have included words like “some” and “many” throughout to indicate that not everyone volunteered this information, but patterns were clear and robust enough to comprise themes. 

Further, the last sentence for the “Inconvenience of accessing services” notes exceptions: “However, a few noted concerns about privacy within their rural communities and noted that, as such, they would like to be able to access VHA healthcare, if eligible (lines 445-447).”

Discussion

The discussion would be more compelling if the findings were integrated and discussed together rather than laid out in separate paragraphs. For example, theme 2 reflected women

lacking important information (theme 1) that might change their perceptions (not knowing that there are women’s clinics or designated women’s health providers).

Response: We agree and have re-written the Discussion to further integrate and synthesize our findings across themes, particularly with respect to the first two themes (lines 452-514). 

How do your findings relate to any previous literature on lack of information on eligibility and services?

Response: We have expanded the literature cited in the discussion to further demonstrate that lack on information on eligibility is a consistent barrier for other Veteran populations in accessing VHA care. 

Revised text, found lines 459-464: “Additionally, consistent with prior studies of women Veterans (41,42), rural women Veterans believed that VHA does not provide healthcare relevant to women Veterans’ unique needs and that VHA is a place for “old White men,” despite significant VHA expansion of healthcare services for women Veterans to include comprehensive women’s health services, maternity care coordination, and provision of specialized training to VHA Women’s Health providers in women Veterans’ healthcare needs (43–46).”

Is there any overlap in findings with qualitative literature among VA users?

Response: Indeed, there are some areas of overlap with other research on Veterans using VA services. We have revised the Discussion to acknowledge these. 

Specifically, in the “Improving communication” section we state, “Additionally, consistent with prior studies of women Veterans, who utilized VHA care (41,42), rural, unenrolled women Veterans believed that VHA does not provide healthcare relevant to women Veterans’ unique needs and that VHA is a place for “old White men.” Despite significant VHA expansion of healthcare services for women Veterans to include comprehensive women’s health services, maternity care coordination, and provision of specialized training to VHA Women’s Health providers in women Veterans’ healthcare needs (43–46) (lines 459-464).”

In the “Inconvenience of accessing VHA services” section we expanded upon the text to show that lack of transportation to healthcare services is a noted barriers for non-VA and VA users alike. “Lack of access and transportation, which have been noted to also impede use of healthcare services for rural Veterans in other studies (52–54), were noted barriers among rural women Veterans to using VHA services in the present study (lines 528-530).”

Whether and how the findings differ for rural women vs. all women Veterans isn’t clear. Are there specific themes or elements of the themes that are unique to rural women Veterans (broadband internet, which is mentioned, seems to be one). Any others?

Response: We agree that there are some similarities in findings regarding women Veterans broadly, as well as some findings that were specific to rural women Veterans, including the internet-related issue. We have revised the Discussion to clarify this, when possible, to do so. 

Regarding technology barriers or lack of broadband internet, you may want to consider discussing VA’s digital divide consult, which is intended to help improve access to broadband,

as well as mobile devices.

Response: We appreciate this suggestion and have added the following statement to the Discussion; “Additionally, it may be particularly important to increase rural women Veterans’ awareness of the VA Digital Divide program, which offers discounts on home internet and phone services or provides a device with internet connectivity for Veterans to engage in telehealth (lines 536-538).”

Reviewer #2:

Line #235: Someone’s name is included in the quote. I believe that should be stricken out with

a parenthetical note substituted to keep everything anonymous.

Response: We apologize for this oversight and very much appreciate the Reviewer bringing this to our attention. We have corrected this and removed the section of the quote with the name included and replaced it with “… (line 317).”

---

## [Decision Letter · Decision Letter 1]

28 Jul 2023

Perspectives of VA Healthcare from Rural Women Veterans not Enrolled in or using VA Healthcare

PONE-D-23-00940R1

Dear Dr. Rohs,

We’re pleased to inform you that your manuscript has been judged scientifically suitable for publication and will be formally accepted for publication once it meets all outstanding technical requirements.

Kind regards,

Mathew Albert Wei Ting Lim

Academic Editor

PLOS ONE

Additional Editor Comments (optional):

Reviewers' comments:

Reviewer's Responses to Questions

**Comments to the Author**

1. If the authors have adequately addressed your comments raised in a previous round of review and you feel that this manuscript is now acceptable for publication, you may indicate that here to bypass the “Comments to the Author” section, enter your conflict of interest statement in the “Confidential to Editor” section, and submit your "Accept" recommendation.

Reviewer #1: (No Response)

Reviewer #2: All comments have been addressed

2. Is the manuscript technically sound, and do the data support the conclusions?

Reviewer #1: Yes

Reviewer #2: Yes

3. Has the statistical analysis been performed appropriately and rigorously? 

Reviewer #1: N/A

Reviewer #2: N/A

4. Have the authors made all data underlying the findings in their manuscript fully available?

Reviewer #1: No

Reviewer #2: Yes

5. Is the manuscript presented in an intelligible fashion and written in standard English?

Reviewer #1: Yes

Reviewer #2: Yes

6. Review Comments to the Author

Reviewer #1: The authors have appropriately addressed concerns from the previous reviews. As a final suggestion, I recommend that in lines 51-55 in the introduction that they provide more information on patient aligned care teams (or use a term more familiar to broader readership, like patient centered medical home) and women veterans program managers, as these are VA-specific terms.

Reviewer #2: (No Response)

7. PLOS authors have the option to publish the peer review history of their article (what does this mean?). If published, this will include your full peer review and any attached files.

Reviewer #1: No

Reviewer #2: No

---

## [Editor Report · Acceptance letter]

4 Aug 2023

PONE-D-23-00940R1 

Perspectives of VA Healthcare from Rural Women Veterans not Enrolled in or using VA Healthcare 

Dear Dr. Rohs:

I'm pleased to inform you that your manuscript has been deemed suitable for publication in PLOS ONE. Congratulations! Your manuscript is now with our production department. 

Kind regards, 

on behalf of

Dr Mathew Albert Wei Ting Lim 

Academic Editor

PLOS ONE